# Willingness-to-Pay of Converting a Centralized Power Generation to a Distributed Power Generation: Estimating the Avoidance Benefits from Electric Power Transmission

Sungwook Yoon

Korea Information Society Development Institute, Jincheon-gun 27872, Chungcheongbuk-do, Republic of Korea; ysungwook@kisdi.re.kr

**Abstract:** Centralized power generation has been widely used for power generation due to its high efficiency, but its transmission and distribution facilities have caused a lot of economic and social costs. The distributed power generation, which produces electricity distributed around consumers without large transmission facilities, has emerged as an alternative. This study aims to derive the social costs and damage avoidance benefits of converting centralized into distributed power generation through willingness-to-pay (WTP) and find determinants that affect them. The economic and social damage caused by centralized power generation facilities is categorized into four types of damage, and the WTP for each type of damage is elicited using various types of quantitative, machine-learning models. Results show that people perceive health damage relief as the greatest benefit of the avoidance of centralized power generation facilities, and it accounted for 29~51% of the total WTP.

**Keywords:** willingness-to-pay; contingent valuation method; centralized power generation; distributed power generation

## 1. Introduction

To date, centralized power generation methods such as bituminous coal thermal and nuclear power generation have played a major role as they have high efficiency. They need large-scale power plants and are located relatively far from major electricity consumption areas due to the ease of selecting power plant sites. Accordingly, several power transmission and distribution facilities are needed to supply electricity to demand sites. However, these facilities have recently caused various social conflicts and costs, such as the Miryang power tower dispute in Korea that has been going on for more than 15 years, since 2008.

As the social cost of such a centralized power generation method becomes an issue, distributed power generation is in the spotlight. Distributed power generation is a method that can be used to supply the power needed to consumers, using small power generation facilities located near them to compensate for the shortcomings of the centralized power supply already in use. By developing such decentralized measures, social problems caused by centralized power generation facilities can be fundamentally resolved. Therefore, replacing centralized with a distributed power generation method is being actively discussed. According to the Ministry of Trade, Industry, and Energy in Korea, the government is also pursuing a policy to increase the proportion of distributed power sources by approximately 5% to 15% by 2035.

However, changing power generation facilities requires a huge budget, and a thorough preparation process must precede the implementation of such public projects. Conducting a feasibility analysis just as a formality can lead to distortion and inefficiency of resource allocation due to wrong decision making, causing great loss to the national economy. Therefore, it is essential to secure validity by strictly calculating the social costs and avoidance benefits of the damage caused by the alternative introduction of distributed power sources.

This study aims to derive the benefits gained by avoiding social costs and damage through the replacement of centralized generation with distributed power generation facilities, hypothetically through willingness-to-pay (WTP), and to find determinants that affect them. The economic and social damage caused by centralized power generation facilities is categorized into four types of damage (i.e., health damage, falling land prices, damaging the surrounding landscape, social conflicts), and the WTP for each type of damage is calculated using various types of quantitative, machine-learning models. The measurement of WTP for avoidance benefit uses the contingent valuation method (CVM), which is commonly applied in evaluating non-market goods, such as public and environmental goods. Moreover, to derive the WTP and its implications according to various individual characteristics, it is modeled through various empirical models.

The remainder of this paper is organized as follows: Section 2 presents the research methodology including the contingent valuation method and the quantitative model used to measure WTP, and the relevant literature. Section 3 summarizes the characteristics, samples, and analysis results of the survey used in the CVM. In Section 4, the representative value of the WTP by members of society is calculated using the results from the previous section and Section 5 summarizes the results based on the aforementioned.

## 2. Research Methodology

This section describes the research method used in this study and the relevant literature. This study uses the contingent valuation method and quantitative models to estimate the amount of intention to pay the avoidance benefit. I estimate the WTP of the total sample by using the dichotomous choice responses, followed by an additional survey on WTP. Probit and Heckman's sample selection models are used to estimate the WTP.

### 2.1. Contingent Valuation Method (CVM)

In the case of goods and services with social values, such as public and environmental goods, their market prices are unavailable because they are not traded in the open market. This complicates the translation of their economic value of increased welfare, from the use of such social goods or services into monetary value. There are two main ways, indirect valuation method and contingent valuation method, to estimate the benefits of social value goods. In the indirect benefit estimation method, the monetary value of the goods can be evaluated indirectly by observing changes in the price or quantity of other market goods in a substitute or supplementary relationship due to the introduction of the target goods. However, if there are no substitutes or complementary goods for the valuation goods, the monetary value cannot be measured via indirect valuation.

Contrary to the indirect benefit estimation method, which measures benefits by observing the characteristics of substitutes or complementary goods measured in the market, the contingent valuation method directly measures consumers' preference by setting a virtual market for the transaction of the evaluated goods, with consumers expressing their WTP for them. In general, face-to-face, mail, or phone interviews are used to gather the opinions of the members of society on the value of the goods under various scenarios with several combined conditions. Under these conditions, respondents provide their ideas about the extent to which they are willing to pay (WTP) for hypothetical changes in evaluated goods.

Based on robust microeconomic theories, the CVM is applicable to goods to which indirect valuation cannot be applied, not to mention those that can be measured using the indirect method [1]. However, as a direct measurement method through surveys, the CVM is susceptible to a range of biases due to the respondents' intention and ability, strategic behavior, and information asymmetry. Thus, the stages of the questionnaire, from preparation to composition and survey design, should be meticulously controlled [2].

### 2.1.1. Theoretical Background

The CVM derives the WTP using Hicks' concepts of 'compensating' and 'equivalent' variation. Compensating variation refers to income adjustment necessary to accept the

changed level of goods, while maintaining the existing level of utility under a given condition. Equivalent variation refers to the adjustment of income necessary to remain at the level of utility after the change.

A compensating surplus can be defined as follows:

$$CS = E(P^0, q^0, Q^0, U^0) - E(P^0, q^i, Q^0, U^0) = Y^0 - Y^i \tag{1}$$

where E is an expenditure function, $P^0$ is the current price of private goods, $Q^0$ is the other level of public good supply, $U^0$ is the current level of utility, and $q^0$ and $q^i$ are the initial and final level of goods to be evaluated, respectively.

Here, $Y^0$ is the current level of expenditure, $Y^1$ is the level of expenditure to maintain the current level of utility when accepting the goods subject to evaluation at the changed level, and the WTP is the difference between $Y^0$ and $Y^1$. Meanwhile, Willig [3] shows that the above function can also be expressed in an equivalent form to the income compensating function. Therefore, when the WTP is used as a measure of benefit, the income compensation function is as follows:

$$WTP(q^i) = f(P^0, q^0, q^i, Q^0, Y^0) \tag{2}$$

Based on the equation above, the change in economic welfare caused by the introduction of goods to be evaluated is expressed in the monetary value of the WTP.

### 2.1.2. Willingness-to-Pay (WTP)

At the core of the CVM lies the creation of a questionnaire that allows the respondents to express their WTP easily. Four types of willingness-to-pay induction methods (direct question approach, auction approach, payment card approach, and dichotomous choice approach) have been devised to induce a more accurate payment environment that reflects the actual preferences of members of society.

The direct question approach involves asking an open-ended question about what is the maximum amount that they are willing to pay for the goods of valuation. Except for the terms of payment, no other subsidiary data is presented. Currently, the direct question method is rarely used for it has been found that participants find it difficult to provide answers about the benefits of the improvement due to the introduction of goods with monetary value [4].

The auction approach is the oldest and is most frequently used to induce WTP, by asking respondents if they are willing to pay a certain amount without asking the amount directly, and raising the offer amount if the answer is "yes", but lowering it if it "no". The advantage of the auction approach is that the more competent the investigator, the higher the WTP. However, it is also known that the amount of WTP is affected by the initial amount [5].

The payment card approach was first introduced by Hanemann [6], in which the investigator presents a series of cards with numbers (payment details for other public goods consumption) to the respondent as auxiliary data, and assisted by these cards, the respondent reveals the amount of WTP. However, the payment details written on the cards should not have much to do with the evaluated goods because if the two are closely related, it has been found that respondents' WTP converges to a value close to the payment card (anchor point bias).

The dichotomous choice approach allows respondents to decide whether or not to buy the goods of valuation at a given price, enquiring from them whether they are willing to pay a preset amount. The WTP is set according to the respondent's answer. Depending on the number of questions asked, this approach is divided into single-bounded and double-bounded dichotomous choice approaches.

In the single-bounded dichotomous choice approach, if the answer to the WTP question is "yes", the WTP is equal to or greater than the amount presented. However, if "no", then it will be equal to or smaller than the amount presented. The double-bounded

dichotomous choice approach specifies the scope of the WTP by raising the offer amount to respondents who answer "yes" to the initial offer, and lowering it to those who answer "no". Compared to the single-bounded dichotomous choice approach, it improves statistical efficiency by drawing more responses from respondents that can be used to estimate the quantitative model on a given number of samples.

### 2.2. Willingness-to-Pay and the Probit Model

Willingness-to-pay ($WTP_i$) of the new goods of respondent $i$ can be expressed as a function of the expenditure difference between the two situations, i.e., where the goods are introduced ($q_i = 1$) and the current situation ($q_i = 0$) without changing the utility level ($u_i$).

$$WTP_i = E(P, q_i = 0, Q, u_i) - E(P, q_i = 1, Q, u_i) = \beta_0 + \beta_1 X_i + \varepsilon_i, \ \varepsilon_i \sim \text{i.i.d.} N(0, \sigma^2) \quad (3)$$

Here, $P$ is the price of private goods, $Q$ is the level of supply of other public goods, $X_i$ is a vector expressing the personal characteristics of respondent $i$, and $\varepsilon_i$ is an error term containing elements that are not observed by the researcher. If dichotomous choice is used to elicit the WTP, and the distribution of the error term is assumed to be a normal distribution, the above-stated model follows the probit model.

### 2.2.1. Willingness-to-Pay and Single-Bounded Dichotomous Choice

The single-bounded dichotomous choice approach is a technique of modeling the amount of WTP by inducing "yes" as an answer if the respondent's own WTP is higher than the amount presented, and "no" if it is lower.

Where $WTP_i$ is the WTP by respondent $i$, $price_i$ is the amount presented to respondent $i$, $X_i$ is the vector of explanatory variables that affect the WTP, and $y_i$ is the indicator variable with 1 and 0, whereby, when respondent $i$ responds with "yes", it is 1 and 0 when "no". The probability is expressed as follows:

$$P_{i1} = P(y_i = 1) = P(WTP_i \geq price_i) = 1 - \Phi\left(\frac{price_i - X_i\beta}{\sigma}\right) \quad (4)$$

$$P_{i2} = P(y_i = 0) = P(WTP_i < price_i) = \Phi\left(\frac{price_i - X_i\beta}{\sigma}\right) \quad (5)$$

Here, $\Phi$ denotes the cumulative probability distribution of the standard normal distribution, and $\sigma$ denotes the standard deviation of the error term. Parameters $\beta$ and $\sigma$ are estimated that maximize the following likelihood function:

$$\ln L = \sum_{i=1}^{n} [y_i \log(P_{i1}) + (1 - y_i) \log(P_{i2})] \quad (6)$$

The goods to be evaluated using CVM have a non-negative utility value because they generally have a public property characteristic. In this case, a log normal distribution limiting the WTP to a positive value, used with the "zero-bid" respondents who chose KRW 0 as the WTP, is excluded from the sample. The amount of WTP for the respondents who answered a non-zero WTP is modeled as follows:

$$\log(WTP_j) = E(P, q_j = 0, Q, u_j) - E(P, q_j = 1, Q, u_j) = \beta_0 + \beta_1 X_j + \varepsilon_j, \ \varepsilon_j \sim \text{i.i.d.} N(0, \sigma^2) \quad (7)$$

The probability according to the respondents' response can be expressed as follows:

$$P_{j1} = P(y_j = 1) = P(WTP_j \geq price_j) = 1 - \Phi\left(\frac{\log(price_j) - X_j\beta}{\sigma}\right) \quad (8)$$

$$P_{j2} = P(y_j = 0) = P(WTP_j < price_j) = \Phi\left(\frac{\log(price_j) - X_j\beta}{\sigma}\right) \quad (9)$$

Using the above probability, the log likelihood function is expressed as follows:

$$\ln L = \sum_{j=1}^{m} \left[ y_j \log(P_{j1}) + (1 - y_j) \log(P_{j2}) \right] \tag{10}$$

### 2.2.2. Willingness-to-Pay and Double-Bounded Dichotomous Choice

The double-bounded dichotomous choice elicits the respondent's WTP by presenting the amount to the respondent twice. The scope of the WTP is specified by raising the offer amount to respondents who answer "yes" to the initial offer and lowering it to respondents who answer "no". The response to the question about the problem of maximizing utility for each amount presented is a combination of "yes" and "no". If respondent $i$'s WTP is greater than the initial offer ($price_{1,i}$), but less than the second ($price_{2,i}$), it is classified as a "yes-no" response. In the survey, twice the amount of the first price ($price_{2,i} = 2 * price_{1,i}$) is offered as the second price to the respondents who answered "yes" at the first question, and half the amount ($price_{2,i} = \frac{1}{2} * price_{1,i}$) is offered as the second price to the respondents who said "no" at the first question.

Since Kriström [7], it has become common to ask additional questions about whether or not respondents are willing to pay even KRW 1 to those who answer "no–no" to the first and second offers. If the answer is no, it is classified as "zero-bid". "Zero-bid" responses include "non-protest response", where changes in public goods have no effect on utility or no ability to pay, and "protest response", which means withholding answers due to distrust of the enforcement entity, lack of information, or antipathy to the survey.

The probability according to the respondents' response can be expressed as follows:

$$P_{i1} = P(y_{1i} = 1, y_{2i} = 1) = P(WTP_i \geq price_{2,i}) = 1 - \Phi\left(\frac{price_{2,i} - X_i\beta}{\sigma}\right) \tag{11}$$

$$P_{i2} = P(y_{1i} = 1, y_{2i} = 0) = P(price_{1,i} \leq WTP_i < price_{2,i}) = \Phi\left(\frac{price_{2,i} - X_i\beta}{\sigma}\right) - \Phi\left(\frac{price_{1,i} - X_i\beta}{\sigma}\right) \tag{12}$$

$$P_{i3} = P(y_{1i} = 0, y_{2i} = 1) = P(price_{2,i} \leq WTP_i < price_{1,i}) = \Phi\left(\frac{price_{1,i} - X_i\beta}{\sigma}\right) - \Phi\left(\frac{price_{2,i} - X_i\beta}{\sigma}\right) \tag{13}$$

$$P_{i4} = P(y_{1i} = 0, y_{2i} = 0, y_{i3} = 1) = P(0 < WTP_i < price_{2,i}) = \Phi\left(\frac{price_{2,i} - X_i\beta}{\sigma}\right) - \Phi\left(\frac{-X_i\beta}{\sigma}\right) \tag{14}$$

$$P_{i5} = P(y_{1i} = 0, y_{2i} = 0, y_{i3} = 0) = P(WTP_i \leq 0) = \Phi\left(\frac{-X_i\beta}{\sigma}\right) \tag{15}$$

where $WTP_i$ denotes the respondent $i$'s willingness-to-pay, $price_{i,1}$ and $price_{i,2}$ as the amount of the first and second offer to respondent $i$, respectively, $y_{i1}$ and $y_{i2}$ denote the response for the first and second offer, and $y_{i3}$ denotes whether he or she responded as "zero-bid".

The log likelihood function can be expressed as follows:

$$\ln L = \sum_{i=1}^{n} \left[ y_{i1}y_{i2} \log(P_{i1}) + y_{i1}(1 - y_{i2}) \log(P_{i2}) + (1 - y_{i1})y_{i2} \log(P_{i3}) + (1 - y_{i1})(1 - y_{i2})y_{i3} \log(P_{i4}) \right.$$
$$\left. + (1 - y_{i1})(1 - y_{i2})(1 - y_{i3}) \log(P_{i5}) \right] \tag{16}$$

If we assume non-negative WTP due to the nature of public goods, a log normal distribution with limiting the WTP to a positive value is used with excluding respondents with a "zero-bid" responses from the sample. The amount of WTP for respondents who respond to non-zero is modeled as follows:

$$\log(WTP_j) = E(P, q_j = 0, Q, u_j) - E(P, q_j = 1, Q, u_j) = \beta_0 + \beta_1 X_j + \varepsilon_j, \ \varepsilon_j \sim \text{i.i.d.N}(0, \sigma^2) \tag{17}$$

The probability according to the respondents' response can be expressed as follows:

$$P_{j1} = P(y_{1j} = 1, y_{2j} = 1) = P(WTP_j \geq price_{2,j}) = 1 - \Phi(\frac{\log(price_{2,j}) - X_j\beta}{\sigma}) \quad (18)$$

$$\begin{aligned} P_{j2} = P(y_{1j} = 1, \quad y_{2j} = 0) &= P(price_{1,j} \leq WTP_j < price_{2,j}) \\ &= \Phi\left(\frac{\log(price_{2,j}) - X_j\beta}{\sigma}\right) - \Phi\left(\frac{\log(price_{1,j}) - X_j\beta}{\sigma}\right) \end{aligned} \quad (19)$$

$$\begin{aligned} P_{j3} = P(y_{1j} = 0, \quad y_{2j} = 1) &= P(price_{2,j} \leq WTP_j < price_{1,j}) \\ &= \Phi\left(\frac{\log(price_{1,j}) - X_j\beta}{\sigma}\right) - \Phi\left(\frac{\log(price_{2,j}) - X_j\beta}{\sigma}\right) \end{aligned} \quad (20)$$

$$P_{j4} = P(y_{1j} = 0, y_{2j} = 0) = P(0 < WTP_j < price_{2,j}) = \Phi\left(\frac{\log(price_{2,j}) - X_j\beta}{\sigma}\right) \quad (21)$$

The log likelihood function can be expressed as follows:

$$\ln L = \sum_{j=1}^{m}\left[y_{j1}y_{j2}\log(P_{j1}) + y_{i1}(1 - y_{i2})\log(P_{i2}) + (1 - y_{i1})y_{i2}\log(P_{i3}) + (1 - y_{i1})(1 - y_{i2})y_{i3}\log(P_{i4})\right] \quad (22)$$

If the WTP is limited to a positive value, "zero-bid" respondents with zero WTP are excluded from the sample space, so their WTP is considered zero and the representative WTP in the entire sample is calculated.

### 2.2.3. Willingness-to-Pay and the Selection Model

When the WTP is limited to a positive value in the double-bounded dichotomous choice, "zero-bid" respondents who mark their WTP as zero are simply excluded from the sample. However, if "zero-bid" occurs systematically among the respondents and there is a statistical correlation between "zero-bid" and WTP, simply removing the "zero-bid" respondents can cause sample selection bias [8,9].

Assuming $Y_i^*$ is the latent "zero-bid" judgment variable of the $i$-th respondent, $Y_i$ is the indicator variable with a value of 1 if the $i$-th respondent expresses WTP more than zero, and $Z_i$ as the explanatory variable of the "zero-bid" judgment. In this case, $Y_i^*$ and $Y_i$ are modeled as follows:

$$Y_i^* = \alpha_0 + \alpha_1 Z_i + v_i, \ v_i \sim i.i.d.N(0,1), \ if \ Y_i^* \geq 0, \ then \ Y_i = 1, \ else \ Y_i = 0 \quad (23)$$

Here, $v_i$ is the error term and assumed to follow a standard normal distribution. If the respondent's latent "zero-bid" judgment variable is more than zero, they respond to the questionnaire ($Y_i = 1$), and if it is less, they respond to "zero-bid". If respondent $i$ is not a "zero-bid" respondent ($Y_i = 1$), the researcher can observe the WTP, and model it as the follows:

$$\log(WTP_i) = \beta_0 + \beta_1 X_i + \varepsilon_i, \ \varepsilon_i \sim i.i.d.N(0, \sigma^2) \ if \ Y_i = 1, \ Y_i^* \geq 0 \quad (24)$$

where $WTP_i$ is respondent to $i$'s WTP, $\varepsilon_i$ is an error term that follows a normal distribution with an average of 0 and a standard deviation of $\sigma$, and $X_i$ is an explanatory variable vector that affects the WTP amount of respondent $i$.

The error terms $(v_i, \ \varepsilon_i)$ of $Y_i^*$ and $\log(WTP_i)$, are modeled with the following bivariate normal distribution.

$$\begin{pmatrix} v_i \\ \varepsilon_i \end{pmatrix} \sim BVN\left(\begin{pmatrix} 0 \\ 0 \end{pmatrix}, \begin{pmatrix} 1 & \rho\sigma \\ \rho\sigma & \rho\sigma^2 \end{pmatrix}\right) \quad (25)$$

Parameter $\rho$ reflects the effects of the unobserved correlates between "zero-bid" judgment and the WTP, and if it has an estimate of $+$, the "zero-bid" judgment and the WTP

amount are in the same direction as that of the unobserved factors. The log likelihood function can be expressed as follows:

$$\ln L = \sum_{i=1}^{n} \begin{matrix} [(1-Y_i)\log(1-\Phi(\alpha_0+\alpha_1 Z_i)) \\ +Y_i\{log\phi\left(\frac{\log(WTP_i)-\beta_0-\beta_1 X_i}{\sigma}\right)-log\sigma \\ +log\Phi\left(\frac{\alpha_0+\alpha_1 Z_i+\rho\frac{(\log(WTP_i)-\beta_0-\beta_1 X_i)}{\sigma}}{\sqrt{1-\rho^2}}\right)\}] \end{matrix} \quad (26)$$

where $\phi$ refers to the probability distribution function of the standard normal distribution, whereas $\Phi$ refers to the cumulative probability distribution function. The full information maximum likelihood estimation (FIML) method was used for parameter estimation.

### 2.3. Literature Review

Willingness-to-pay estimation using the contingent valuation method has been widely used in cost-benefit analysis of public policy in the field of energy and environment sector. Longo et al. [10] elicited the WTP for the hypothetical policy of renewable energy. In a study analyzing WTP related to a new German plan for transforming their energy provision system ("Energiewende"), they found obvious dissimilarity between the households' attitude towards renewable energy technologies and their WTP amount for green electricity [11]. The benefits of avoiding the damage from large-scale transmission facilities are estimated in Kim et al. [12] and they revealed that people are willing to pay 33% more of their current electricity bill to avoid that damage.

In the WTP research related to power generation, one of the main interests was the destruction of the landscape. Ladenburg and Dubgaard [13] estimated WTP for reducing the visual disamenities from future offshore wind farms. In Mirasgedis et al. [14], WTP related to landscape destruction by large-scale wind power generation was measured, and they found that 43% of samples are willing to contribute financially in order to prevent landscape damage. In terms of overhead power transmission, a research study revealed that the social benefits of avoiding the negative impacts on landscape exceed the cost of undergrounding transmission cables [15].

The relationship between energy generation and health is also a topic of interest for researchers to measure the benefits of public policy. Itoka et al. [16] estimated WTP for the reduction of health damage caused by fossil fuel versus power generation and found that WTP for avoiding health problems related to nuclear power generation is about 60 times larger than the WTP related to fuel generation. Pandey and Nathwani [17] presented the life quality index (LQI) as a tool for the assessment of health risk reduction in industrial installations including energy generation facilities and LNG terminals.

In a study related to the introduction of a public policy in environment sector, Longo et al. [18] measured the WTP of climate change mitigation program in Basque County, Spain. Xiong et al. [19] measured the WTP to improve the ecological environment of the Ganjiang River basin in China. Albernini and Krupnick [20] compared the WTP and the cost of illness (COI) of the respiratory symptoms associated with air pollution and concluded that WTP is about 1.61 to 2.26 times greater than the COI.

Many WTP-related studies have tried to identify internal and external factors that influence the amount of WTP. In Xiong et al. [19], demographic factors including education level, work type, and household annual disposable income affect the amount of WTP, but the effect of value recognition to WTP is at least marginal. Supplementary data explaining the ancillary benefit also had a significant effect on the amount of WTP [18]. In a study related to the installation of nuclear power plants, the distance from the facilities was a significant factor explaining the amount of WTP. The degree of acceptance of green energy technology also affected the WTP [11].

## 3. Empirical Analysis

### 3.1. Questionnaire

The questionnaire employed in this study consisted of: (1) definition, status, and understanding of centralized and distributed power plants; (2) attitude/sentiment measurement of social problems regarding centralized power plants (transmission grid facilities); (3) measurement of WTP; and (4) question regarding the respondents' individual characteristics. In terms of definition, status, and understanding of the two types of facilities, the characteristics and disadvantages of the centralized power generation facilities, the related social issues, and the avoidance benefits of the distributed power generation facilities were explained based on four damage categories, i.e., health damage, falling land prices, damaging the surrounding landscape, and social conflicts. To measure the attitude toward social problems of centralized facilities, respondents were asked how well they perceived the social problems caused by the transmission network facilities of centralized power generation facilities, and how serious they thought the problems were, based on the four damage categories described above. To measure the WTP, respondents were presented with the average electricity bill per household (about KRW 25,000, about USD 20) in 2017 as a reference, and were asked whether they are willing to make additional payments for the installation of distributed power generation facilities. A total of five questions were asked, and the first to the fourth questions consisted of dichotomous questions, giving them an option of answering "yes/no" to the amount presented, and the fifth question about maximum WTP was asked only to those respondents whose WTP was greater than zero. The fifth question is similar to the direct question method, but it can elicit a more accurate amount in that it helps respondents to convert the benefits of improvement from the introduction of valuation goods into a monetary value.

As for the individual characteristics of the respondents, information about household heads/members, average monthly income (pre-tax income), average monthly electricity bills, residential areas, and gender was collected.

### 3.2. Sample Design and Data Collection

A specialized survey agency conducted the survey, and the sample consisted of 1500 randomly selected people, aged between 19 to 65 years, who were picked nationwide in Korea. The survey was conducted through one-on-one individual phone interviews. It is necessary to set realistic payment methods to reduce response bias. The WTP was measured by asking how much of the cost of electricity they could additionally pay from the average monthly electricity bill of KRW 25,000 (about USD 20), if power generation facilities are replaced from centralized to decentralized. The initial offer amounts were set as KRW 2500, KRW 5000, and KRW 7500, which were 10%, 20%, and 30% of the average monthly electricity bill, respectively. Table 1 describes the sample characteristics.

Table 2 summarizes the WTP responses for each damage type. The number of respondents answering "yes" to the first offer in health damage avoidance is the highest. This means that respondents recognize the harmful effects to health as the biggest concern in various types of damage caused by concentrated power generation facilities. However, the majority of people answered that they were not willing to pay the first offer for the benefit of avoiding social conflict. This means that they did not perceive social conflicts as the most serious issue among the various types of damage, or if they did, they had no intention to avoid social conflict caused by centralized power generation by paying out of their own pockets, even if it is serious. The higher the first amount offered for each type of damage, the lower the number of respondents who expressed their WTP, and the greater the proportion of those who refused to pay it. Meanwhile, the share of "zero-bid" to the total response is not significantly related to the first offer amount. The "zero-bid" share is found to be the highest in "social conflict" and the lowest in "health damage" avoidance, respectively.

**Table 1.** Characteristics of the sample.

| | | Results | |
|---|---|---|---|
| | | **Count** | **%** |
| Total | | 1500 | 100.0 |
| Gender | Male | 766 | 51.1 |
| | Female | 734 | 48.9 |
| Age | 19~29 | 315 | 21.0 |
| | 30~39 | 311 | 20.7 |
| | 40~49 | 363 | 24.2 |
| | 50~59 | 364 | 24.3 |
| | 60~64 | 147 | 9.8 |
| Household Owner/Household Member | Household Owner/Partner | 980 | 65.3 |
| | Household Member | 520 | 34.7 |
| Electricity Bill (Monthly, KRW 1000) | ~10 | 58 | 3.9 |
| | 10~60 | 1065 | 71.0 |
| | 60~120 | 311 | 20.7 |
| | 120~170 | 43 | 2.9 |
| | 170~ | 23 | 1.5 |
| Region | Seoul | 295 | 19.7 |
| | Gyeonggi/Incheon | 471 | 31.4 |
| | Chungcheong | 155 | 10.3 |
| | Honam | 141 | 9.4 |
| | Daegu, Gyeongbuk | 144 | 9.6 |
| | Busan, Ulsan, Gyeongnam | 230 | 15.3 |
| | Gangwon, Jeju | 64 | 4.3 |
| Average Monthly Income (KRW 1M) | ~300 | 421 | 28.1 |
| | 300~500 | 522 | 34.8 |
| | 500~700 | 341 | 22.7 |
| | 700~ | 216 | 14.4 |
| Initial Offer (KRW) | 2500 | 505 | 33.7 |
| | 5000 | 491 | 32.7 |
| | 7500 | 504 | 33.6 |

**Table 2.** WTP response by damage type.

| | KRW | Yes (%) Y-Y | Y-N | Total | No (%) N-Y | N-N | Zero-Bid | Total | Total |
|---|---|---|---|---|---|---|---|---|---|
| Health Damage | 2500 | 172 (34.1) | 129 (25.5) | 301 (59.6) | 51 (10.1) | 35 (6.9) | 118 (23.4) | 204 (40.4) | 505 (100.0) |
| | 5000 | 102 (20.8) | 130 (26.5) | 232 (47.3) | 81 (16.5) | 61 (12.4) | 117 (23.8) | 259 (52.7) | 491 (100.0) |
| | 7500 | 100 (19.8) | 103 (20.4) | 203 (40.3) | 86 (17.1) | 90 (17.9) | 125 (24.8) | 301 (59.7) | 504 (100.0) |
| Land Prices Falling | 2500 | 128 (25.3) | 116 (23.0) | 244 (48.3) | 59 (11.7) | 41 (8.1) | 161 (31.9) | 261 (51.7) | 505 (100.0) |
| | 5000 | 78 (15.9) | 114 (23.2) | 192 (39.1) | 57 (11.6) | 53 (10.8) | 189 (38.5) | 299 (60.9) | 491 (100.0) |
| | 7500 | 88 (17.5) | 80 (15.9) | 168 (33.3) | 74 (14.7) | 87 (17.3) | 175 (34.7) | 336 (66.7) | 504 (100.0) |
| Surrounding Landscape Damage | 2500 | 130 (25.7) | 126 (25.0) | 256 (50.7) | 49 (9.7) | 49 (9.7) | 151 (29.9) | 249 (49.3) | 505 (100.0) |
| | 5000 | 76 (15.5) | 128 (26.1) | 204 (41.5) | 60 (12.2) | 66 (13.4) | 161 (32.8) | 287 (58.5) | 491 (100.0) |
| | 7500 | 78 (15.5) | 86 (17.1) | 164 (32.5) | 76 (15.1) | 90 (17.9) | 174 (34.5) | 340 (67.5) | 504 (100.0) |
| Social Conflict | 2500 | 111 (22.0) | 107 (21.2) | 218 (43.2) | 33 (6.5) | 44 (8.7) | 210 (41.6) | 287 (56.8) | 505 (100.0) |
| | 5000 | 56 (11.4) | 102 (20.8) | 158 (32.2) | 50 (10.2) | 64 (13.0) | 219 (44.6) | 333 (67.8) | 491 (100.0) |
| | 7500 | 62 (12.3) | 87 (17.3) | 149 (29.6) | 54 (10.7) | 85 (16.9) | 216 (42.9) | 355 (70.4) | 504 (100.0) |

### 3.3. Estimation Result

This section estimates the WTP of the respondents, using various models, based on the responses collected through the survey. Table 3 summarizes the empirical model used to estimate the WTP.

**Table 3.** Empirical model used for estimating WTP.

| | Model | Response Data | Distribution of WTP | Range of WTP | "Zero-Bid" |
|---|---|---|---|---|---|
| 1-1 | Dichotomous Choice | Single-Bounded | Normal Distribution | $(-\infty, \infty)$ | X |
| 1-2 | Dichotomous Choice | Single-Bounded | Log Normal Distribution | $(0, \infty)$ | Excluded from Sample |
| 2-1 | Dichotomous Choice | Double-Bounded | Normal Distribution | $(-\infty, \infty)$ | $(-)$ WTP |
| 2-2 | Dichotomous Choice | Double-Bounded | Log Normal Distribution | $(0, \infty)$ | Excluded from Sample |
| 3 | Selection | Amount Willing to Pay | Log Normal Distribution | $(0, \infty)$ | Decision Variable |

Models 1-1 to 2-2 estimate the WTP using the dichotomous choice model, and Model 3 uses Heckman's selection model. Models 1-1 and 1-2 estimate the model using only the response for the first offer of the positive response data for each damage type, whereas Models 2-1 and 2-2 estimate the WTP function using the double positive response. In Model 3, WTP is estimated using responses of respondents with zero or more (non-zero-bid), and the "zero-bid" judgment variable is also modeled to handle selection bias. The WTP in Models 1-1 and 2-1 theoretically ranges from $-\infty$ to $\infty$. In Models 1-2 and 2-2, the WTP is modeled with log normal distribution to reflect the non-negative characteristics of willingness to pay for public goods, and respondents with "zero-bid" are excluded. In this case, the representative value of the WTP is adjusted by the proportion of "zero-bid" respondents.

#### 3.3.1. Results of the Single-Bounded Dichotomous Choice

Table 4 summarizes the estimation results of Model 1-1 about the WTP with a normal distribution. The younger the age, the more the people are willing to pay for damage by type of intensive development. Based on the benefit of avoiding health problems, those in their 20s, 30s, 40s, and 50s were ready to pay KRW 6609, KRW 4448.8, KRW 3959.8, and KRW 2610.1 more, respectively, than those in their 60s. Parameter estimates related to age were significant in the WTP for all types of damage.

The heads of the household showed greater avoidance benefits than the members, showing a greater WTP of about KRW 2050.6 more than the members to avoid health problems. In other types of damage, whether the respondent is the household head was not significant. The attitude toward understanding the facility and its problems was partially significant. The higher the understanding about the facility, the more consideration was given to the seriousness of the damage, and the greater was the WTP. Meanwhile, the standard deviation of each WTP is greater in the category of "conflict" than that of "health", because "zero-bid" occurs more in the responses of the "conflict" damage type.

Table 5 summarizes the estimation results of Model 1-2, in which participants who did not respond with "zero-bid" were extracted for when the WTP was modeled with a log-normal distribution. As in Model 1-1, it can be confirmed with significant positive estimates that the younger the age, the higher the WTP for each type. Unlike Model 1-1, in 1-2, the amount of WTP by those in their 50s and 60s was not statistically different. Regarding questions about facility understanding or damage recognition, estimates were generally not significant. In Model 1-1, the standard deviation of "social conflict" was

greater than that of other types of damage. However, the results of Model 1-2 show that the standard deviation of "land prices" was greater than that of "social conflict". Since this was analyzed after excluding "zero-bid" respondents in Model 1-2 and the highest proportion of "zero-bid" was observed in the "social conflict", the volatility related to WTP was more offset in "social conflict" than other damage types. The overall statistical significance of the parameter decreased because the observations in Model 1-2 decreased by as little as 360 and as much as 645, compared to Model 1-1.

**Table 4.** Estimation results of the single-bounded dichotomous choice approach (1-1).

| | Variable | Value | Health Damage | Land Price Falling | Surrounding Landscape Damage | Social Conflict |
|---|---|---|---|---|---|---|
| Individual Characteristics | Constant | | −11,337 *** | −12,500 ** | −10,183 ** | −18,085 *** |
| | Sex | | | | Not Valid | |
| | Age (Standard: 60) | 20 s | 6609.1 *** | 6313.4 *** | 5587 *** | 6459.5 *** |
| | | 30 s | 4448.8 *** | 4805.2 *** | 4115.3 *** | 6048.6 *** |
| | | 40 s | 3959.8 *** | 3972.2 ** | 3780.6 *** | 6016.8 *** |
| | | 50 s | 2610.1 ** | 3540.7 ** | 3397.5 ** | 3932.1 ** |
| | Owner/Member | Owner | 2050.6 ** | 1673 | 654.2 | 1748.6 |
| | Income | | | | Not significant | |
| | Electricity Bill | | | | Not significant | |
| | Region | | | | Not significant | |
| Understanding of Facilities | Facilities 1 | 1 | 1661.7 | 3376 | 1250.7 | 2930.2 |
| | | 2 | 1533 * | 2405 ** | 200.5 | 2888.1 ** |
| | Facilities 2 | | | | Not significant | |
| Attitude/Sentiment | Damage 1 | 1 | 4703.2 *** | 3686.7 ** | 2543.7 * | 3036.4 |
| | | 2 | 2961.2 ** | 1042.6 | 463.9 | 1285.9 |
| | Damage 2 | 1 | 5406.7 ** | 3135.9 | 5383.5 ** | 7279.3 * |
| | | 2 | 4591.3 * | 2541.4 | 4972.3 ** | 5617.1 |
| | | 3 | 2032.8 | 947 | 4951.6 ** | 1072.9 |
| | Standard Error | | 9448.3 *** | 11,663 *** | 9982.1 *** | 12,804 *** |
| | Observations | | 1500 | 1500 | 1500 | 1500 |
| | Log-likelihood | | −949.6 | −935.3 | −952.2 | −899.8 |
| | AIC | | 1991.2 | 1962.5 | 1996.3 | 1891.6 |

(Facility 1): asking whether you know about the main power generation method (centralized power generation), with three answer choices of 1 (I know well), 2 (I know a little), and 3 (I do not know at all). (Facility 2): asking whether you know about the distributed power generation, with choices the same as (Facility 1). (Problem 1): asking whether you know about each type of damage, with choices the same as (Facility 1). (Problem 2): asking about the severity of each type of damage, with answer choices of 1 (very serious), 2 (somewhat serious), 3 (not very serious), and 4 (not serious at all). *** $p$-value < 0.01, ** $p$-value < 0.05, * $p$-value < 0.1.

**Table 5.** Estimation results of single-bounded dichotomous choice approach (1-2).

| | Variable | Value | Health Damage | Land Price Falling | Surrounding Landscape Damage | Social Conflict |
|---|---|---|---|---|---|---|
| Individual Characteristics | Constant | | 7.69 *** | 7.25 *** | 7.25 *** | 6.58 *** |
| | Sex | | | | Not significant | |
| | Age (Standard: 60) | 20 s | 0.8 *** | 0.9 ** | 0.94 *** | 0.65 * |
| | | 30 s | 0.51 ** | 0.63 * | 0.61 ** | 0.71 ** |
| | | 40 s | 0.47 ** | 0.3 | 0.43 | 0.55 * |
| | | 50 s | 0.26 | 0.37 | 0.46 | 0.51 |
| | Owner/Member | Owner | | | Not significant | |
| | Income | | | | Not significant | |
| | Electricity Bill | | | | Not significant | |
| | Region | | | | Not significant | |

**Table 5.** *Cont.*

| | Variable | Value | Health Damage | Land Price Falling | Surrounding Landscape Damage | Social Conflict |
|---|---|---|---|---|---|---|
| Understanding of Facilities | Facilities 1 | 1 | 0.34 | 0.65 | 0.28 | 0.85 ** |
| | | 2 | 0.16 | 0.28 | −0.08 | 0.32 |
| | Facilities 2 | | | Not significant | | |
| Attitude/Sentiment | Damage 1 | 1 | 0.46 * | 0.6 ** | 0.1 | 0.59 * |
| | | 2 | 0.33 | 0.22 | −0.12 | 0.63 * |
| | Damage 2 | 1 | 0.08 | 0.09 | 1.06 ** | 1.2 * |
| | | 2 | −0.27 | 0.03 | 0.79 * | 0.92 |
| | | 3 | −0.45 | −0.31 | 0.95 * | 0.6 |
| | Standard Error | | 1.45 *** | 1.86 *** | 1.73 *** | 1.71 *** |
| | Observation | | 1140 | 975 | 1014 | 855 |
| | Log-likelihood | | −664.2 | −582.8 | −617.9 | −510.7 |
| | AIC | | 1420.4 | 1257.5 | 1327.7 | 1113.3 |

(Facility 1): asking whether you know about the main power generation method (centralized power generation), with three answer choices of 1 (I know well), 2 (I know a little), and 3 (I do not know at all). (Facility 2): asking whether you know about the distributed power generation, with choices the same as (Facility 1). (Problem 1): asking whether you know about each type of damage, with choices the same as (Facility 1). (Problem 2): asking about the severity of each type of damage, with answer choices of 1 (very serious), 2 (somewhat serious), 3 (not very serious), and 4 (not serious at all). *** $p$-value < 0.01, ** $p$-value < 0.05, * $p$-value < 0.1

### 3.3.2. Results of the Double-Bounded Dichotomous Choice

Table 6 summarizes the estimation results of Model 2-1, which models the amount of WTP as a normal distribution. As with the results in the single-bounded dichotomous choice model (Model 1-1), the lower the age group, the higher the WTP for each type of damage, and the higher the age group, the lower the WTP. Based on "health damage", people in their 20s, 30s, and 40s are willing to pay approximately KRW 3865.3, KRW 2375.2, and KRW 1558 more, respectively, than those in their 60s. The household owner's status created a significant difference in the WTP regarding "health damage" and "social conflict". Unlike the single-bounded dichotomous choice model, a parameter for the first offer is added in an empirical model, and the higher the amount, the higher the respondent's potential WTP. In case of the "social conflict", if the first offer was KRW 7500, the WTP was approximately KRW 2027.8 higher than if it was KRW 2500. In cases of "health damage" and "surrounding landscape damage", the more serious the respondents considered the damage caused by the centralized power generation facilities, the higher the WTP. Participants who said that the health damage was very serious were willing to pay approximately KRW 4025.8 more than those who said it was not.

**Table 6.** Double-bounded dichotomous choice method (2-1) estimation results.

| | Variable | Value | Health Damage | Land Prices Falling | Surrounding Landscape Damage | Social Conflict |
|---|---|---|---|---|---|---|
| | Constant | | −5946 *** | −5799 *** | −5996 *** | −6424 ** |
| | Sex | | | Not significant | | |
| Individual Characteristics | Age | 20 s | 3865.3 *** | 3661 *** | 2807 *** | 2789.2 *** |
| | (Standard:60 s) | 30 s | 2375.2 *** | 2876.2 *** | 2470.4 *** | 2276.8 ** |
| | | 40 s | 1931.2 ** | 2564.8 *** | 2274.6 *** | 2535 *** |
| | | 50 s | 1558 ** | 2147.7 ** | 1779.8 ** | 1384.5 |
| | Owner/Member | Owner | 1136.3 ** | 789.5 | 701.5 | 1180.2 * |
| | Income | | | | Not significant | |
| | Electricity Bill | | | | Not significant | |
| | Region | | | | Not significant | |
| | Initial Offer | KRW 5000 | 1078.5 ** | 405.9 | 844.3 * | 299.6 |
| | (Baseline: KRW 2500) | KRW 7500 | 2632.6 *** | 2378.9 *** | 1965.9 *** | 2027.8 *** |

**Table 6.** *Cont.*

| | Variable | Value | Health Damage | Land Prices Falling | Surrounding Landscape Damage | Social Conflict |
|---|---|---|---|---|---|---|
| Understanding of Facilities | Facilities 1 | 1 | 1638.9 * | 2212.5 ** | 1307.1 | 1558.2 |
| | | 2 | 1315.7 ** | 1631.6 *** | 859.4 | 2100.1 *** |
| | Facilities 2 | | | | Not significant | |
| Attitude/Sentiment | Damage 1 | 1 | 3235 *** | 1919.7 *** | 1850.1 ** | 498.8 |
| | | 2 | 1916.6 *** | 531 | 526.7 | −398.7 |
| | Damage 2 | 1 | 4025.8 *** | 2196.2 | 2999.3 ** | 2429.6 |
| | | 2 | 3833.7 *** | 1564.6 | 2723.8 ** | 1544.2 |
| | | 3 | 2037.4 | 885 | 2147.6 * | −639.7 |
| | Standard Error | | 6933.5 *** | 7776.3 *** | 7264.5 *** | 8003.3 *** |
| | Observation | | 1500 | 1500 | 1500 | 1500 |
| | Log-likelihood | | −2314.1 | −2282.1 | −2316.3 | −2149.4 |
| | AIC | | 4724.1 | 4660.1 | 4728.5 | 4394.8 |

(Facility 1): asking whether you know about the main power generation method (centralized power generation), with three answer choices of 1 (I know well), 2 (I know a little), and 3 (I do not know at all). (Facility 2): asking whether you know about the distributed power generation, with choices the same as (Facility 1). (Problem 1): asking whether you know about each type of damage, with choices the same as (Facility 1). (Problem 2): asking about the severity of each type of damage, with answer choices of 1 (very serious), 2 (somewhat serious), 3 (not very serious), and 4 (not serious at all). *** $p$-value < 0.01, ** $p$-value < 0.05, * $p$-value < 0.1.

Table 7 summarizes the estimation results of Model 2-2 in which only participants who did not respond with "zero-bid" were extracted for which the WTP is modeled with a log-normal distribution. The WTP of those people in their 20s was statistically and significantly higher than those in their 60s for each type of damage except for "social conflict". The difference in WTP between people of other ages (30s, 40s, and 50s) and those in their 60s was insignificant. Even if excluding "zero-bid" respondents, the statistically significant estimates demonstrate that the initial offer still results in a significant difference in the amount of WTP.

**Table 7.** Double-bounded dichotomous choice method (2-2) estimation results.

| | Variable | Value | Health Damage | Land Price Falling | Surrounding Landscape Damage | Social Conflict |
|---|---|---|---|---|---|---|
| | Constant | | 8.19 *** | 7.84 *** | 7.53 *** | 7.32 *** |
| | Sex | | | | Not significant | |
| Individual Characteristics | Age | 20 s | 0.31 ** | 0.39 ** | 0.31 ** | 0.2 |
| | (Baseline: 60 s) | 30 s | 0.11 | 0.24 * | 0.2 | 0.16 |
| | | 40 s | 0.12 | 0.11 | 0.15 | 0.16 |
| | | 50 s | 0.01 | 0.15 | 0.06 | 0.09 |
| | Owner/Member | Owner | | | Not significant | |
| | Income | | | | Not significant | |
| | Electricity Bill | | | | Not significant | |
| | Region | | | | Not significant | |
| | Initial Offer | KRW 5000 | 0.31 *** | −[1] | 0.4 *** | 0.28 *** |
| | (Baseline: KRW 2500) | KRW 7500 | 0.54 *** | 0.42 *** | 0.62 *** | 0.6 *** |
| Understanding of Facilities | Facilities 1 | 1 | 0.21 | 0.36 ** | 0.14 | 0.46 *** |
| | | 2 | 0.12 | 0.18 * | 0.04 | 0.24 ** |
| | Facilities 2 | | | | Not significant | |
| | Damage 1 | 1 | 0.27 ** | 0.24 ** | 0.02 | 0.22 |
| | | 2 | 0.14 | 0.12 | −0.11 | 0.26 * |
| Attitude/Sentiment | Damage 2 | 1 | −0.13 | −0.04 | 0.49 *** | 0.58 ** |
| | | 2 | −0.27 | −0.15 | 0.32 | 0.43 |
| | | 3 | −0.33 | −0.33 | 0.38 * | 0.45 |
| | Standard Error | | 0.87 *** | 0.94 *** | 0.89 *** | 0.92 *** |
| | Observation | | 1140 | 975 | 1014 | 855 |
| | Log-likelihood | | −1499.1 | −1323.15 | −1371.85 | −1171.85 |
| | AIC | | 2998.2 | 2646.3 | 2743.7 | 2343.7 |

(Facility 1): asking whether you know about the main power generation method (centralized power generation), with three answer choices of 1 (I know well), 2 (I know a little), and 3 (I do not know at all). (Facility 2): asking whether you know about the distributed power generation, with choices the same as (Facility 1). (Problem 1): asking whether you know about each type of damage, with choices the same as (Facility 1). (Problem 2): asking about the severity of each type of damage, with answer choices of 1 (very serious), 2 (somewhat serious), 3 (not very serious), and 4 (not serious at all). *** $p$-value < 0.01, ** $p$-value < 0.05, * $p$-value < 0.1. [1] Excluded from explanatory variables due to a parameter identification problem that occurred during the elimination of "zero-bid" respondents.

### 3.3.3. Results of the Selection Model

This section describes the results of the Heckman selection model, which handles sample selection bias by simultaneously modeling the amount of WTP and "zero-bid" behavior. Table 8 summarizes the parameter estimates for the "zero-bid" discriminant part of the selection model.

**Table 8.** Selection model: "zero-bid" discrimination.

| | Variable | Value | Health Damage | Land Price Falling | Surrounding Landscape Damage | Social Conflict |
|---|---|---|---|---|---|---|
| Individual Characteristics | Constant | | −0.29 | −0.78 | −0.2 | −0.56 |
| | Sex | Male | −0.11 | −0.19 *** | −0.1 | −0.14 ** |
| | Age | 20 s | 0.62 *** | 0.46 *** | 0.38 *** | 0.35 ** |
| | (Baseline: 60 s) | 30 s | 0.45 *** | 0.38 *** | 0.34 ** | 0.28 ** |
| | | 40 s | 0.31 ** | 0.43 *** | 0.36 *** | 0.36 *** |
| | | 50 s | 0.33 ** | 0.34 *** | 0.31 ** | 0.16 |
| Understanding of Facilities | Initial Offer | | | | Not significant | |
| | Facilities 1 | | | | Not significant | |
| | Facilities 2 | | | | Not significant | |
| Attitude/Sentiment | Damage 1 | 1 | 0.55 *** | 0.16 | 0.35 *** | 0.01 |
| | | 2 | 0.33 *** | 0.02 | 0.14 | −0.19 |
| | Damage 2 | 1 | 0.74 *** | 0.41 ** | 0.26 | 0.18 |
| | | 2 | 0.89 *** | 0.36 ** | 0.36 * | 0.14 |
| | | 3 | 0.47 * | 0.33 * | 0.23 | −0.22 |

Through the selection model, it can be estimated who will select "zero-bid", along with the individual's WTP. (Problem 1): asking whether you know about each type of damage, with choices the same as (Facility 1). (Problem 2): asking about the severity of each type of damage, with answer choices of 1 (very serious), 2 (somewhat serious), 3 (not very serious), and 4 (not serious at all). *** *p*-value < 0.01, ** *p*-value < 0.05, * *p*-value < 0.1.

All damage types had significantly positive estimates for those in their 20s to 50s, and this means that people in their 20–50s did not opt for "zero-bid" more than those in their 60s. Therefore, younger people had a higher WTP or actively responded to the survey. In addition, people who responded to the impact of centralized power generation facilities on health damage and serious land price decline have actively participated in the survey.

Table 9 summarizes the estimation results of the amount of WTP using the selection model. Except for damage related to "social conflict", the WTP by people in their 20s was greater than those in other age groups. For all types of damages, the estimate of the parameter of the first offer had a significantly negative value. This means that the lower the initial amount, the lower the respondent's WTP. On the other hand, the correlation coefficient was not significant in all types, which means that the unobserved correlates between the WTP and "zero-bid" behavior were not significant.

Models 1-1 and 2-1 had estimated the WTP by putting all of them in samples without considering whether "zero-bid" was significantly influenced by age groups, but Models 1-2 and 2-2 that had excluded "zero-bid" showed a significant difference only in the 20s age group. Based on the results of estimating "zero-bid" using a selection model, people in their 20s to 50s expressed their WTP more actively than those in their 60s, and did not opt for "zero-bid". Taken together, Models 1-1 and 2-1 showed that non-zero-bid respondents in their 30s–50s were not willing to pay more than those in their 60s, but due to a decrease in "zero-bid" owing to active payment responses from people in their 30s–50s, the WTP seemed higher.

However, the first offered amount was found to be non-significant in the determination of "zero-bid" in the selection model, but it was significant in the estimation of the WTP. Therefore, the first offer was irrelevant to the "zero-bid" behavior, but it affected the non-zero-bidders' amount of WTP.

**Table 9.** Selection model: estimation result of WTP.

| | Variable | Value | Health Damage | Land Prices Falling | Surrounding Landscape Damage | Social Conflict |
|---|---|---|---|---|---|---|
| | Constant | | 9.2 *** | 8.82 *** | 8.66 *** | 8.09 *** |
| Individual Characteristics | Male | | | | Not Valid | |
| | Age (Baseline: 60 s) | 20 s | 0.4 *** | 0.37 ** | 0.39 *** | 0.28 |
| | | 30 s | 0.18 | 0.17 | 0.22 | 0.25 |
| | | 40 s | 0.2 ** | 0.12 | 0.19 | 0.29 |
| | | 50 s | 0.09 | 0.09 | 0.07 | 0.14 |
| | Owner/Member | Owner | | | Not Valid | |
| | Income | | | | Not Valid | |
| | Electricity Bill | | | | Not Valid | |
| | Region | | | | Not Valid | |
| | Initial Offer (Baseline: KRW 7500) | KRW 2500 | −0.45 *** | −0.58 *** | −0.51 *** | −0.51 *** |
| | | KRW 5000 | −0.21 *** | −0.19 ** | −0.16 ** | −0.25 ** |
| Understanding of Facilities | Facilities 1 | | | | Not Valid | |
| | Facilities 2 | | | | Not Valid | |
| Attitude/Sentiment | Damage 1 | 1 | 0.36 *** | 0.34 *** | 0.05 | 0.24 |
| | | 2 | 0.22 * | 0.17 * | −0.15 | 0.19 |
| | Damage 2 | | | | Not Valid | |
| | Standard Error | | 0.99 *** | 0.96 *** | 0.99 *** | 1.1 *** |
| | Correlation Coefficient | | 0.06 | 0.04 | 0.1 | −0.02 |
| | Log-likelihood | | −2364 | −2265 | −2329 | −2270 |
| | AIC | | 4920 | 4722 | 4851 | 4731 |

(Facility 1): asking whether you know about the main power generation method (centralized power generation), with three answer choices of 1 (I know well), 2 (I know a little), and 3 (I do not know at all). (Facility 2): asking whether you know about the distributed power generation, with choices the same as (Facility 1). (Problem 1): asking whether you know about each type of damage, with choices the same as (Facility 1). (Problem 2): asking about the severity of each type of damage, with answer choices of 1 (very serious), 2 (somewhat serious), 3 (not very serious), and 4 (not serious at all). *** *p*-value < 0.01, ** *p*-value < 0.05, * *p*-value < 0.1.

### 3.4. Discussion

The parameter estimates differed slightly from each other in direction and magnitude. The reason is as follows: first, the response data used in each model is different. Models 1-1 and 1-2 only used the response to the first offered amount of the questionnaire, but Models 2-1 and 2-2 used both the response to the first and second offered amount. In addition, Model 3 used data describing the specific amount of payment provided directly by the respondent, rather than responses from a dichotomous choice approach. Since the types and levels of responses used in each model are different, the estimates of the model and the WTP thereafter are bound to be different.

Using different samples also makes each model-specific estimate different. Since Models 1-2 and 2-2 assume the non-negative WTP, respondents with "zero-bid" were excluded, and the sample was reduced by at least 360 and at most 645, hence, indicating a possibility of the samples' characteristics being changed.

The different probability distributions used for empirical models also cause differences in estimation results. In the case of Models 1-2, 2-2, and 3, the WTP follows the log normal distribution due to the non-negative WTP, but the remaining models use the normal distribution. Log transformation of the random variables changes the meaning of the estimate of each variable, thereby, changing the value.

Three types of models used to estimate the amount of WTP (single- and double-bounded choice, and selection model) have different characteristics, advantages, and disadvantages. In the double-bounded choice, the range of the WTP is determined using two responses according to the offered amount for each respondent, estimating the influence of each variable and the WTP by each of them. Therefore, statistical efficiency in-

creases by using more contingent valuation response data compared to the single-bounded dichotomous choice model. In other words, it can be estimated with fewer samples than the single-bounded dichotomous choice model in measuring the impact of explanatory variables and predicting the WTP. However, a starting point bias in which the estimates of the WTP change according to the initial offered amount occurs. In Models 2-1 and 2-2, the effect of the first offered amount is significant, which means that the respondent's potential WTP changes according to the first offered amount. If the respondent's stated preferences match their actual WTP, the potential payment amount should not change under the given conditions, and thus, this starting point bias should not occur. In addition, response bias exists, in which potential WTP generated only using the response to the first or second offer differ from each other [21,22].

Unlike the double-bounded dichotomous choice model, the single-bounded dichotomous choice model is free from the starting point and response bias. However, the available response data used in estimation is reduced by half compared to the above-mentioned response, which means that twice as many samples are needed to estimate the parameters with the same accuracy.

The selection model deals with "zero-bid" using multiple explanatory variables, preventing sample selection bias that may occur when "zero-bid" occurs systematically among respondents, and unobserved correlates exist between "zero-bid" response propensity and WTP. In addition, the effect of explanatory variables can be divided into the effect on the respondent's WTP and "zero-bid". However, since the "zero-bid" response is further modeled, it results in a doubling of the number of parameters, which reduces statistical efficiency.

In the case of "zero-bid", the behavior of respondents who have no intention to pay due to the introduction of new public goods or lack of ability to pay, or refuse to pay by not answering the questions even though the reservation price for public goods is not zero, must be distinguished to accurately measure the benefits and WTP. An additional protest response filtering is required to distinguish these various respondents, but as no additional questions had been asked to distinguish these responses, they had not been separately distinguished within "zero-bid".

## 4. Representative Value of the Willingness-to-Pay

I elicited the representative expected WTP due to the avoidance benefit of the centralized power grid damage through the conversion of respondents to distributed power sources. Table 10 summarizes the expected WTP function using the coefficients of the parameters estimated in Section 3:

**Table 10.** Selection model: expected WTP function.

| Model | Expected WTP Function | |
|---|---|---|
| Model 1-1 | $E[\widehat{WTP_i}] = \widehat{\beta_{1-1,0}} + \widehat{\beta_{1-1,1}}X_i$ | (27) |
| Model 1-2 | $E[\widehat{WTP_i}] = \exp\left(\widehat{\beta_{1-2,0}} + \widehat{\beta_{1-2,1}}X_i\right)$ | (28) |
| Model 2-1 | $E[\widehat{WTP_i}] = \widehat{\beta_{2-1,0}} + \widehat{\beta_{2-1,1}}X_i$ | (29) |
| Model 2-2 | $E[\widehat{WTP_i}] = \exp\left(\widehat{\beta_{2-2,0}} + \widehat{\beta_{2-2,1}}X_i\right)$ | (30) |
| Model 3 | $\widehat{Y_i^*} = \widehat{\alpha_0} + \widehat{\alpha_1}X_i,$ $E[\widehat{WTP_i}] = \exp\left(\widehat{\beta_{3,0}} + \widehat{\beta_{3,1}}X_i + E[\varepsilon_i\mid\widehat{Y_i^*}]\right) * I\left(\widehat{Y_i^*} \geq 0\right)$ | (31) |

where $E[\widehat{WTP_i}]$ denotes the expected WTP of respondent $i$, $X_i$ denotes a group of individual characteristic explanatory variables, $\widehat{Y_i^*}$ denotes the predictive latent "zero-bid" judgment of model 3, $\widehat{\beta_{1-1,0}}, \ldots, \widehat{\beta_{3,0}}$ denote the estimates of the constant term of WTP for each model, $\widehat{\beta_{1-1,1}}, \ldots, \widehat{\beta_{3,1}}$ denote the estimates of the explanatory variable in the WTP for each model, $\widehat{\alpha_0}$ and $\widehat{\alpha_1}$ are the estimates of the "zero-bid" judgment, and $I(\cdot)$ is the indicator function, which is assigned 1 if the value in parentheses is true, and if not, 0. Using the specifications above, the expected WTP ($E[\widehat{WTP_i}]$) of respondents are calculated. Thereafter, the median value of the respondents' expected WTP is used as a representative value. In Models 1-1 and 2-1, some respondents' WTP may have negative values, and their WTP values are considered as zero ("zero-bid").

Since "zero-bid" respondents are excluded in model 1-2 and 2-2 due to non-negative WTP, the representative value of the WTP must be adjusted by the number of excluded samples. The representative value is adjusted by multiplying the median value of the WTP with the proportion of "non-zero-bid" respondents. In Model 3, the expected WTP is multiplied by the expected probability of individual i's being a "non-zero-bid" respondent. If respondent *i* has a greater tendency to "zero-bid", the expected WTP converges to 0. The reason that the median WTP is set as the representative value instead of the average WTP is because Models 1-2 and 2-2 are limited to the non-negative WTP, resulting in a right-skewed distribution, and it makes the average value of expected WTP less representative [12]. Therefore, the median, and not the average, is used as the representative WTP.

Table 11 summarizes the WTP for each damage type by avoidance benefit, by model. The total indicates the sum of the four types of WTP. Since the above four damage factors cannot be regarded as mutually exclusive and all-inclusive elements of the total damage of the centralized power generation facility transmission network felt by the respondent, the simple sum calculated above cannot be interpreted as the benefit of avoiding the total damage. Therefore, the total sum should be used for simple reference only. The total amount of WTP is the lowest and highest in Model 1-1 and 1-2, respectively, recording KRW 9767 and KRW 21,923. Since the type of responses used to estimate the WTP differs for each model, the various amounts of WTP have been estimated for each model. Model 2-1 elicits a relatively small amount of WTP, unlike Models 1-2 and 2-2. Since Models 1-1 and 2-1 estimate the WTP by including "zero-bid" samples, these "zero-bid" samples cause a downward bias in parameters due to the large proportion of these samples. The amount of WTP for each type of damage is the highest and lowest for the benefits of avoiding health damage and social conflict, respectively.

**Table 11.** WTP by type of avoidance benefits (unit: KRW).

|  | Model 1-1 | Model 1-2 | Model 2-1 | Model 2-2 | Model 3 |
|---|---|---|---|---|---|
| Health Damage | 4960 | 6305 | 4914 | 4621 | 4376 |
| Land Price Falling | 1911 | 5438 | 2860 | 3618 | 3946 |
| Surrounding Landscape Damage | 2868 | 5415 | 3194 | 3659 | 3721 |
| Social Conflict | 28 | 4765 | 1286 | 2944 | 3138 |
| Total | 9767 | 21,923 | 12,254 | 14,843 | 15,182 |

## 5. Conclusions

Currently, Korea mainly depends on centralized power generation, using nuclear power or coal to supply power, which accounts for over 70% of its total energy production. Owing to the issues of safety and environmental hazards, these facilities are usually located away from the metropolitan area and large cities, whereto electricity is transmitted over large-scale power transmission grids. However, these power grid facilities incur various social costs due to health hazards and opposition from residents. To reduce these conflicts and costs, lively discussions have been underway to install distributed power generation facilities such as solar power generation and cogeneration facilities, and to increase the proportion thereof. Against this backdrop, the government has set the goal of expanding distributed power sources, comprising 5% of total power today to 15% of total generation by 2035. The social costs and damage avoidance benefits of converting centralized power generation facilities that require large-scale power transmission to distributed power sources were surveyed through a contingent valuation method of 1500 randomly selected people, picked nationwide in Korea. In 2017, the average electricity bill per household (approximately KRW 25,000) was presented, and additional payment due to the installation of distributed power generation facilities was requested. By this, it was possible to measure the WTP by avoidance benefits for each of the four types of damage, and the total differed from KRW 9767 to KRW 21,923, depending on the statistical model used in the analysis. Comparison of WTP confirms that people perceive health damage relief as the

greatest benefit of the avoidance of centralized power generation facilities. The WTP for health damage relief accounted for 29~51% of the total WTP. The feasibility of carrying out public works, such as changes in power generation facilities, should be considered before undertaking the project because they involve an economic cost to stakeholders as well as difficult-to-quantify benefits and social problem solutions. Additionally, as solution of social problems in public sectors is not traded through the price mechanism of the market, quantifying and measuring the monetary value involves many difficulties. This study is meaningful as it quantifies the solution of social problems by converting the social benefits of changing the types of power generation facilities into a monetary value, and can be used as a reference for economic evaluation of future decentralized power expansion policies.

A few limitations of this study should be noted. First, this study collected 1500 respondents with various characteristics, but the number of respondents is relatively small given that Korea's population is about 50 million. It is necessary to collect more samples in order to elicit precise amounts of WTP. Second, this study elicits WTP based on the perceived risk and the benefits of converting centralized power generation to distributed power generation, and consumer perceptions related to the costs of installation, operation, and energy efficiency of distributed generation are not reflected in WTP. Cost-related factors in the operating process of distributed power generation are also factors that can affect consumers' WTP, so it is necessary to specify and handle them in follow-up studies. Lastly, the survey included questions asking how much the respondents were aware of the social problems caused by transmission network facilities, but not a question asking about the level of understanding of the difference between the two types of the power generation. The degree of respondents' understanding could affect the amount of WTP, and this relationship can be investigated by incorporating responses to additional questions about level of understanding into the model. I hope that results of this study encourage further investigation on these issues.

**Funding:** This work was supported by the KISDI Global ICT Trend Analysis, SK Networks and PwC.

**Institutional Review Board Statement:** Not applicable.

**Informed Consent Statement:** Not applicable.

**Data Availability Statement:** The data presented in this study are available on request from the author.

**Conflicts of Interest:** The author declares no conflict of interest.

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
