# Peer review of "Willingness-to-Pay of Converting a Centralized Power Generation to a Distributed Power Generation: Estimating the Avoidance Benefits from Electric Power Transmission"

_sustainability, doi:10.3390/su15064949_

Round 1

Reviewer 1 Report

This study aims to derive the social costs and damage avoidance benefits of converting centralized into distributed power generation facilities through  willingness-to-pay  (WTP)  and  find determinants that affect  them.

This problem is an age-old question of energy scientists: whether to transport coal to a distant power station or to build a power station near the mine and distribute energy from it. In the case of nuclear power plants, which are large-scale producers of energy, there is therefore nothing left but to distribute the energy from them over long distances. For nuclear power plants with smaller power, this is a current problem in the solution stage.

There are 25 equations in the article, I think it would be appropriate to number them for greater clarity. Alogoritmus (on lines 253, 254) could perhaps be separated from the rest of the text by an empty line, just like the equations, so that they do not get lost in it.

Author Response

Comments and Suggestions for Authors

→I am very appreciative for all the constructive comments and suggestions received from you and the rest of the review team. As explained in my responses in this document, I have addressed all of these points and revised the paper accordingly. Again, we thank you and the rest of the review team for all your input.

This study aims to derive the social costs and damage avoidance benefits of converting centralized into distributed power generation facilities through willingness-to-pay (WTP) and find determinants that affect them.

→Thank you very much for the careful reading and for the nice summary of our paper.

This problem is an age-old question of energy scientists: whether to transport coal to a distant power station or to build a power station near the mine and distribute energy from it. In the case of nuclear power plants, which are large-scale producers of energy, there is therefore nothing left but to distribute the energy from them over long distances. For nuclear power plants with smaller power, this is a current problem in the solution stage.

→Thank you very much for raising issue. I agree that methodological discussions on converting centralized power generation to distributed power generation have been extensively researched through the efforts of energy scientists. However, in Korea, policy goals have recently been set to promote the spread of distributed power in earnest, and in order to supply and utilize distributed power, it is necessary to measure the benefits and WTP for distributed power from the perspective of energy consumers. A research related to the establishment of an electricity pricing system for distributed power generation is being conducted in Korea recently [1], and result of this study can be used as a reference for constructing pricing policy from the consumer’s prospective.

There are 25 equations in the article, I think it would be appropriate to number them for greater clarity. Alogoritmus (on lines 253, 254) could perhaps be separated from the rest of the text by an empty line, just like the equations, so that they do not get lost in it.

→Thank you for this comment. In the revised manuscript, I numbered all equations to help readers understand better. I also expressed equations on lines 252 and 253 in one line like other equations to make it easier to understand. Please refer to page 6 in the revised manuscript.

Reference

[1]       A study on transmission and distribution rate system for effective distribution and utilization of distributed power system (2022), Korea Energy Economics Institute.  

Reviewer 2 Report

This is an interesting study to identify the Willingness to pay for distributed power generation facilities as opposed to centralized power generation facilities. Although, this is a broader topic that needs to take into account several aspects, the authors focus on the role of consumers. 

  • The studies made by the authors include 1500 participants with lot of diversity. The reviewer feels the number of participants are less compared to the population. 

  • Other factors such as the comparison of power losses between distributed and centralized generation, their respective costs of installation, operation, willingness of the consumer to participate in the distributed generation, etc., also play a role but are not considered. The reviewer is just mentioning this for the sake of completeness of the topic and is not suggesting the authors to consider this in their study. A brief discussion around this would be helpful as the current title is very generic. 

  • There are several sentence formation issues and grammatical errors in the manuscript. I would encourage the authors to send the manuscript to a technical editor before their next submission. 

Author Response

Comments and Suggestions for Authors

→I am very appreciative for all the constructive comments and allowing me to resubmit the paper. In the revised manuscript, I have carefully addressed your comments and revised the paper accordingly. Again, I am very appreciative of your comment and believe that it has helped me to produce a much stronger manuscript.

This is an interesting study to identify the Willingness to pay for distributed power generation facilities as opposed to centralized power generation facilities. Although, this is a broader topic that needs to take into account several aspects, the authors focus on the role of consumers. 

→Thank you very much for the careful reading and for the nice summary of our paper.

  • The studies made by the authors include 1500 participants with lot of diversity. The reviewer feels the number of participants are less compared to the population. 

→Thank you for this comment. Due to research funding constraints, I was forced to collect data from a limited number of participants. In the revised manuscript, I have highlighted the limitation of relatively small samples and added this issue as a future research. Please refer to page 20 on the revised manuscript.

  • Other factors such as the comparison of power losses between distributed and centralized generation, their respective costs of installation, operation, willingness of the consumer to participate in the distributed generation, etc., also play a role but are not considered. The reviewer is just mentioning this for the sake of completeness of the topic and is not suggesting the authors to consider this in their study. A brief discussion around this would be helpful as the current title is very generic. 

→Thank you for this comment. I agree that the issues related to the installation, operation, and efficiency of distributed power sources are not reflected, and only the perceived benefits from the consumers’ point of view are considered. In the revised manuscript, I have highlighted the limitation that this study focuses on the perceived benefits of energy consumers and added this issue as a future research. Please refer to page 20 on the revised manuscript.

  • There are several sentence formation issues and grammatical errors in the manuscript. I would encourage the authors to send the manuscript to a technical editor before their next submission. 

→Thank you for your comment. In the revised manuscript, sentences with grammatical errors were corrected with the help of an English editing specialist that had previously been entrusted with English proofreading.  

Reviewer 3 Report

This paper presents the benefits of converting centralized power generation to distributed power generation through willingness-to-pay (WTP). A survey has been performed to evaluate the WTP. The following are my comments.

1.     The equations are confusing because they do not have equation numbers. The author should include equation numbers for each equation.

2.     Table 1 is not clear; the author should change the format of the table.

3.    In Table 1, are the average monthly income (KRW) values correct?

4.    The conducted survey did not include the understanding of the people about the difference between centralized power generation and distributed power generation. Is this having no impact on their responses? 

Author Response

This paper presents the benefits of converting centralized power generation to distributed power generation through willingness-to-pay (WTP). A survey has been performed to evaluate the WTP. The following are my comments.

→I am very appreciative for all the constructive comments and allowing me to resubmit the paper. In the revised manuscript, I have addressed your comments and revised the paper accordingly. Again, thank you and the rest of the review team for all your input.

  1. The equations are confusing because they do not have equation numbers. The author should include equation numbers for each equation.

→Thank you for this comment. In the revised manuscript, I numbered all equations to help readers understand better.

  1. Table 1 is not clear; the author should change the format of the table.

→Thank you for pointing this out. I have changed the format of the table 1 so that the reader can better understand the characteristics of the sample. Please refer pages 8-9 in revised manuscript.

  1.   In Table 1, are the average monthly income (KRW) values correct?

→Thank you for pointing this out. I have modified the unit of average monthly income from KRW to 1M KRW. I am sorry for the confusion with incorrect units. Please refer pages 8-9 in revised manuscript.

  1.   The conducted survey did not include the understanding of the people about the difference between centralized power generation and distributed power generation. Is this having no impact on their responses? 

→Thank you very much for raising this issue. The survey included questions asking how much the respondents were aware of the social problems caused by transmission network facilities, but not a question asking about the level of understanding of the difference between the two types of the power generation. Although the questionnaire described both types of power generations and social problems caused by a centralized power generation, the degree of respondents’ understanding could affect the amount of WTP. We have highlighted this issue as a limitation of the study and added as a future research. Please refer page 20 in revised manuscript.
